# The Use of Fertilizers and Pesticides in Wheat Production in the Main European Countries

Valentina Constanta Tudor [1], Paula Stoicea [1,*], Irina-Adriana Chiurciu [1], Elena Soare [1],
Adina Magdalena Iorga [1], Toma Adrian Dinu [1], Livia David [1], Marius Mihai Micu [1], Dragos Ion Smedescu [1]
and Eduard Alexandru Dumitru [2]

[1]  Faculty of Management and Rural Development, University of Agronomic Sciences and Veterinary Medicine,
    010961 Bucharest, Romania
[2]  Office for Rural Development, Research Institute for Agriculture Economy and Rural Development,
    010961 Bucharest, Romania
*  Correspondence: stoicea.paula@managusamv.ro

**Abstract:** The aspects highlighted in this paper refer to the application of fertilizers and plant protection products to the wheat crop, and their impact on total production, competitiveness in export markets, price optimization as a result of increased harvest quality, and not least row, ensuring farmers' incomes. The present analysis concerned the areas cultivated with wheat, and the productions achieved in relation to the quantities of fertilizers and protection products used in wheat cultivation, in Romania, Germany, Spain, France, Italy, Hungary, Poland and the UK. The analysis highlighted that, the average productions are influenced both by the technology used, by the fertilization and treatments applied, as well as by other synergistic factors that intervene in wheat crops. In countries such as Spain and Italy, the correlation between the amounts of N (nitrogen), $P_2O_5$ (phosphorus), $K_2O$ (potassium) and the average production of wheat is insignificant, but in Romania and Hungary, there is a strong, direct connection between the number of fertilizers used and the average wheat crop yield. In Romania and Hungary, there is a very significant connection, but the production link average with pesticides is negative, while in Germany, the correlation is direct and quite intense for all types of fertilizers used, while the use of pesticides has a lower influence on average wheat production. In this study, we also applied a multiple regression model; in which the dependent variable was the average yield of wheat per ha and the independent variables were the average quantity of pesticides and fertilizers used. We used the "t Stat" values for each coefficient to determine whether the coefficient is equal to zero; where a high value of "t Stat", greater than 2; indicates the fact that the coefficient is significantly different from the value zero. The "*p*-value" for each coefficient indicates the probability that the coefficient has the value zero. A value below the significance level of 5% results in a coefficient significantly different from zero and with an impact on the dependent variable. The use of fertilizers on wheat crops has a synergistic effect as shown by the multiple linear regression analysis which demonstrated a strong direct relationship, particularly with the amount of N, $K_2O$ and $P_2O_5$ applied.

**Keywords:** areas; yield; wheat; nitrogen; phosphorus; potassium; plant protection; EU

## 1. Introduction

Cereals have always occupied an important place in the world food economy because cereal grains constitute a very large proportion of the basic food for the world's population. Cereal production represents only a part of the agricultural production obtained at the global level, which definitely contributes to the establishment of food security [1]. An important aspect is represented by the fact that conventional agricultural systems were an alternative, especially if one takes into account the fact that the numerical growth of the population was accompanied by the increase in food needs. In this context, the grain production achieved at the global level has come to cover part of the consumption needs of

the population [2]. Cereal grains have numerous uses, one of the most important being as a raw material for various industries that contribute significantly to the economic success of many countries. These grains, when ground and processed into products such as bread, semolina, and pasta, are a crucial part of human nutrition and form a staple food for much of the world. They are also an important component of animal feed and serve as a raw material in the production of products such as alcohol, beer, starch, dextrin, and the glucose industry. The "cereals" category includes the following: wheat, rye, barley, oats, triticale, millet, rice, sorghum and corn. According to specialized studies, the mentioned cereals have many characteristics that give them a special value, directly contributing to their importance for human activity, but also for the survival of the species [3]. Cereals are grown all over the globe but in different proportions. For example, wheat culture is particularly well-known and appreciated. It should be specified that there is no uniformity for the wheat crop at the global level, as it is being cultivated in various pedoclimatic conditions, but the final goal is unique, namely, providing food for the population [4].

Regarding wheat, it should be specified that it is the most important cereal crop that is cultivated and from which bread is prepared for approximately 40% of the world's population. In addition to the mentioned, one more essential aspect must be specified, namely, the fact that wheat grains are used in animal feed because they possess a series of proteins, mineral salts and lipids [5]. Due to the fact that wheat is used in food for humans and animals, in various industries, farmers are constantly interested in obtaining it in the appropriate quantity both for consumption and marketing [6]. Wheat production depends on numerous production factors, such as pedoclimatic conditions, cultivated variety, cultivation technology, irrigation level, etc. [7,8]. In the last decades, all of humanity has witnessed, along with specialists, a series of unprecedented climate changes. In this context, the main players in agriculture are forced to find a series of viable solutions, on the one hand, for producer-consumers, and on the other hand, for the preservation of biodiversity [9]. The nutrient content of the soil can support sustainable, high-quality wheat crop production [10–12]. Proper nutrition and the use of local varieties adapted to the specific agro-pedo-climatic conditions are among the factors that contribute to optimal yield in the wheat crop [13,14]. Complex fertilizers that are administered in autumn must have an N:P ratio in favor of $P_2O_5$ or equal, while if complex fertilizers are administered in spring, they must have an N:P ratio in favor of N [3,15]. Sustained application of chemical fertilizers leads to an increase in yields of common wheat and spelt that can double or even triple. It is important to remember that the goal of culture technologies applied to sustainable agriculture is to ensure that agrotechnical, agrochemical and phytosanitary practices applied to the wheat crop do not negatively impact soil fertility and structure and protect the environment as much as possible.

The rates of N fertilizers recommended for application to wheat crops, for medium to high production yields, are 60 kg/ha until the appearance of the first stem node; March–April, before the buds appear, administer 40 kg N/ha [3]. In the European countries studied, the vegetation conditions are suitable for wheat cultivation and the optimal times for performing agrotechnical, agrochemical, and phytosanitary practices are determined based on the specific pedoclimatic conditions of each year. Diagnosing the state of vegetation of the wheat crop is recommended to be carried out on the basis of analyzes carried out before spring fertilizing and targeting N-mineral at a depth of 0–90 cm, in order to establish the N reserve at the arrival of spring and plant density/$m^2$ related to the number of germinated grains. To reduce ammonia evaporation or N loss, the second application of fertilizers is recommended to be carried out in cold and wet weather [10,12]. The minimum yield is the lowest production obtained when one of the weather factors is at its lowest level. The maximum yield is the highest production obtained when all weather factors are at their optimal levels.

N, the main nutrient element, influences the vegetative development of plants; it lacks leads to reduced quantities of total production and poor quality.

$P_2O_5$ is one of the essential macro elements for wheat cultivation. Its application determines the stimulation of root development, increases tolerance to wintering and thermal and water stress, and contributes decisively to the fruiting process.

Certain studies have shown that the levels of $K_2O$ and the time of their application have substantially influenced the yield. In order to obtain high yields, it becomes necessary to administer $K_2O$ on all types of soil; the dose is 40–80 kg $K_2O$/ha [3].

The use of pesticides is important for increasing the quantity and quality of wheat production and protecting the food and health of the population. Pesticides help to control pests and diseases or their vectors, as well as reduce spoilage during storage. Weeds, diseases, insects, and other pests can reduce wheat crop yields, and the use of pesticides helps to minimize these losses. In the early stages of crop development, the use of herbicides to control weed infestations is essential for achieving optimal grain yield and desired economic benefits. [16–19] However, pesticides can also pose a threat to human health and the environment, and European farmers are encouraged to reduce their use and adopt organic and integrated crop systems, though the impact of these changes on crop yields is not yet fully understood [9,20,21].

Unfortunately, the effects produced by climate change affect the productivity of agricultural crops, and also jeopardize the provision of food security, given the current geopolitical context [22]. However, it is necessary to ensure a balance in order to obtain sufficient agricultural production and to protect the environment, but this is difficult to achieve in the context of decreases in the number of fertilizers and pesticides applied in agriculture [23]. Recent studies indicate an openness of farmers in terms of adapting to these requirements, without affecting their profitability at the farm level [24].

Currently, the agricultural sector requires increased attention not only because it provides the necessary agricultural products but also because it plays a significant role in the world economy as a primary sector which by the year 2050 must respond to a major challenge, namely, the doubling of agricultural production, simultaneously with the reduction of pesticides used [25,26]. With a continually growing population, it is necessary to increase food production, which can be achieved by combining the use of fertilizers with culture technologies adapted to the specific pedoclimatic conditions of each country.

## 2. Materials and Methods

The research carried out in the work is based on descriptive statistics indicators for each country. The indicators determined for the years 2010 and 2019 are as follows: the arithmetic mean, the standard deviation, and the coefficient of variation of the amounts of fertilizers and the associated productions for n = 10 years. The existence, meaning, and intensity of the link between the production and the number of fertilizers for each country during the 10 years were studied using the Pearson linear correlation coefficient denoted "r". The more "r" has a positive value, the closer to 1, the stronger and more direct the connection. When the value of "r" approaches -1, the correlation is strong and inverse [27]. The value of "r" close to zero indicates no correlation or a very weak correlation. For a better interpretation of the results, we also considered the critical values of the r coefficient depending on the number of degrees of freedom. Through the linear regression model $Y = ax + b$, where x = the average quantity of fertilizer, being the independent variable, and Y = the average production of wheat is the dependent variable, we estimated the proportion in which the factor quantity of fertilizer influences the average production of wheat for the sample of eight countries studied being given by the coefficient of determination $R^2$ calculated for N, $P_2O_5$, $K_2O$ and pesticides. Through the multiple linear regression model, $Y = a_0 + a_1X_1 + a_2X_2 + a_3X_3$, where $X_1, X_2, X_3$ = the average amount of each fertilizer, Y = the average wheat production (Y = the dependent variable depending on $X_1, X_2, X_3$), we have estimated the proportion in which each factor (amount of fertilizer) influences the average wheat production for the sample of 8 countries studied being given by the determination coefficient $R^2$ calculated for N, $P_2O_5$, $K_2O$, F test and *p*-value. We also studied the interactive input of pesticides and the three fertilizers on the average wheat production based on the

regression model $Y = a_0 + a_1X_1 + a_2X_2 + a_3X_3 + a_4X_4$. The coefficient of an independent variable in a regression equation represents the estimated effect that this variable has on the dependent variable, with all other variables being held constant. If the coefficient for an independent variable is significantly different from zero (as determined by the t-statistic and *p*-value), it means that the variable has a significant impact on the dependent variable.

To carry out the work, the authors studied a series of bibliographic materials that are mentioned in the References section and used statistical data provided by Eurostat [28] and Faostat [29].

This study analyzed data from 2010 to 2019 from specified sources to examine the evolution and multi-year averages of wheat crop area and production, as well as the amounts of fertilizers and pesticides used, in the main wheat-producing countries, according to the share of production obtained in these countries in the total EU wheat production in 2019: France (20.93%), Germany (12.92%), Poland (10.51%), Romania (9.05%), UK (7.60%), Spain (6.92%), Hungary (4.10%) and Italy (2.22%) [28].

## 3. Results and Discussions

According to Eurostat, the total area cultivated with common wheat and spelt (*Triticum aestivum* ssp. *spelta*) in the European Union varied between 2010 and 2019. Spelt is a species of wheat cultivated since the Bronze Age, known for its resistance to frost and diseases and for its flour which is rich in gluten [30]. The lowest area cultivated with common wheat and spelt was recorded in 2018 (23,018.82 thousand ha), and the biggest was 24,419.29 thousand ha (2014). In the European Union, in 2019, the area cultivated with common wheat and spelt increased by 175.87 thousand hectares compared to 2010. One possible reason for this is climate change.

The multi-year average for the total area cultivated with common wheat and spelt in the countries studied varied between 547.58 thousand ha and 4924 thousand ha (Figure 1).

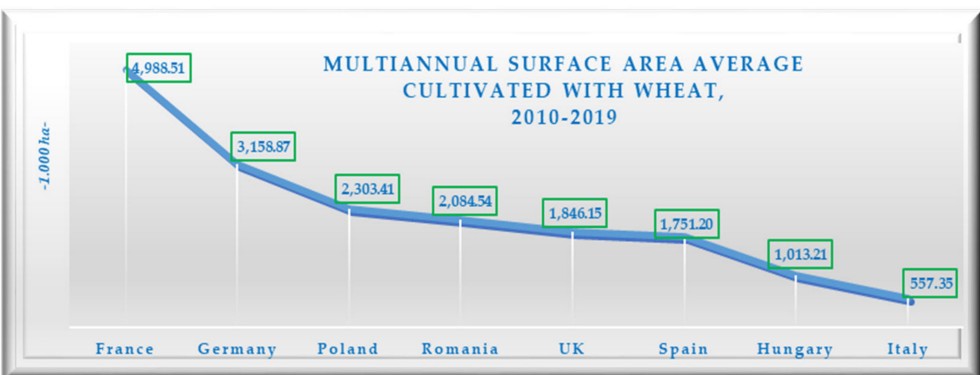

**Figure 1.** Common wheat and spelt—multiannual average for the total area, in the main growing states in the EU. (Source: Eurostat, 8 November 2021 and own processing).

The largest increases in the total area cultivated with common wheat and spelt in the European Union were recorded in 2014, compared to 2013 "+4.33%". The most significant decrease in the total area cultivated with common wheat and spelt was in 2012, compared to 2011 "−1.84%", due to climate change (Figure 2).

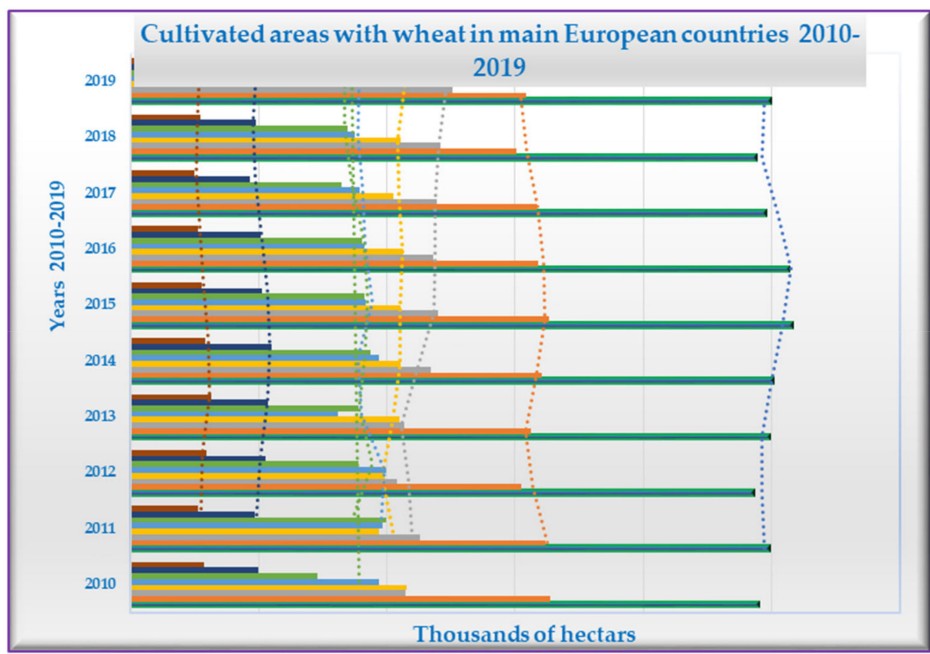

**Figure 2.** Total area cultivated with common wheat and spelt. (Source: Eurostat, 8 November 2021 and own processing).

In Romania, 2105.54 thousand ha represented the multi-annual average for the total area cultivated with common wheat and spelt, which occupied the 4th place in the hierarchy of the countries studied and compared to France, Romania had a total area cultivated with common wheat and spelt of 42.24%.

France. In 2019, 20.92% of the wheat-cultivated area in the European Union was dedicated to growing common wheat and spelt in France. From 2010 to 2019, France had the largest area dedicated to growing these grains (see Table 1). The highest cultivation area was recorded in 2015 at 5161.39 thousand hectares, while the lowest was 4880.21 thousand hectares in 2018, and in 2019, in France, increased by 2.04% (100.23 thousand hectares) compared to 2010. In 2015, there was a 6.24% increase compared to 2012.

**Table 1.** The standard deviation and the coefficient of variation for the multiannual average amount of N were used to obtain one ton of wheat (source: www.fao.org and own processing).

| Country | N fertilizers (a.n. N) | | |
|---|---|---|---|
| | Multiannual Average of the Amount of N Used— kg a.n. N/ton of Wheat | Deviation Standard | The Coefficient of Variation |
| Germany | 134.42 | 12.95 | 9.63 |
| Spain | 57.81 | 5.61 | 9.70 |
| France | 110.80 | 5.24 | 4.73 |
| Italy | 62.54 | 4.92 | 7.86 |
| Hungary | 78.93 | 12.88 | 16.32 |
| Poland | 96.24 | 6.16 | 6.41 |
| Romania | 38.65 | 6.98 | 18.07 |
| UK | 168.69 | 4.71 | 2.79 |

Regarding Poland, in 2019 the total area cultivated with common wheat and spelt represented 10.51% of the area of the European Union, respectively 2511.33 thousand ha

and the largest area increase was recorded in 2011 compared to 2010 (+5.47%), and the most significant decrease was in 2012 compared to 2011 (−8.04%).

In 2019, Romania cultivated 9.05% of the total area cultivated with common wheat and spelt in the European Union and the largest area was 2162.64 thousand ha (2019), and the lowest was 1942.33 thousand ha (2011). The most significant decrease was in 2011 compared to 2010 (−9.66). The trend oscillated slightly during the analysis period.

Regarding the UK, we note that in 2019, this country cultivated 7.60% of the total area of this crop existing at the level of the European Union, and in 2014, compared to 2013, the area cultivated with wheat increased (+19.87%), but it decreased in 2013 compared to 2012 (−18.93%), noting an oscillating trend during the analyzed period.

In 2011, Spain cultivated wheat on 1994.65 thousand hectares, a 36.64% increase compared to the previous year, making up 5.35% of the area dedicated to growing common wheat and spelt in the European Union.

In 2019, Hungary cultivated 4.09% of the total area cultivated with common wheat and spelt in the European Union, and the largest area cultivated was 1098.23 thousand ha (2014), and the smallest was 965.62 thousand ha (2011). The largest increase in the total area cultivated with common wheat and spelt in Hungary was recorded in 2012 compared to 2011 (+9.54%), and the most significant decrease was in 2011 compared to 2019 (−1.33%).

In Italy, it was found that the area cultivated with common wheat and spelt represented 2.22% of the area cultivated in the European Union with this crop, and in 2012, the wheat crop area had the most significant increase compared to the previous year, i.e., 11.73%.

The next part of the study presents the evolution of total and average production for common wheat and spelt crops in the specified countries over the analyzed period.

The multi-year average for the total production of common wheat and spelt in the countries studied varied between 2994.3 thousand tons and 34,790.95 thousand tons. For Romania, the multiannual average for the total production of common wheat and spelt during the analyzed period was 8161.96 thousand tons, and it took 5th place in the ranking of the countries studied. This country held 0.9% of the multi-year average recorded by France and ranked first in total production of common wheat and spelt. The multi-year average for average production/ha of common wheat and spelt in the countries studied varied between 7.85 tons/ha to 3.55 tons/ha.

The total production realized from 2010 to 2019 in the main European countries included in the study is shown in Figure 3.

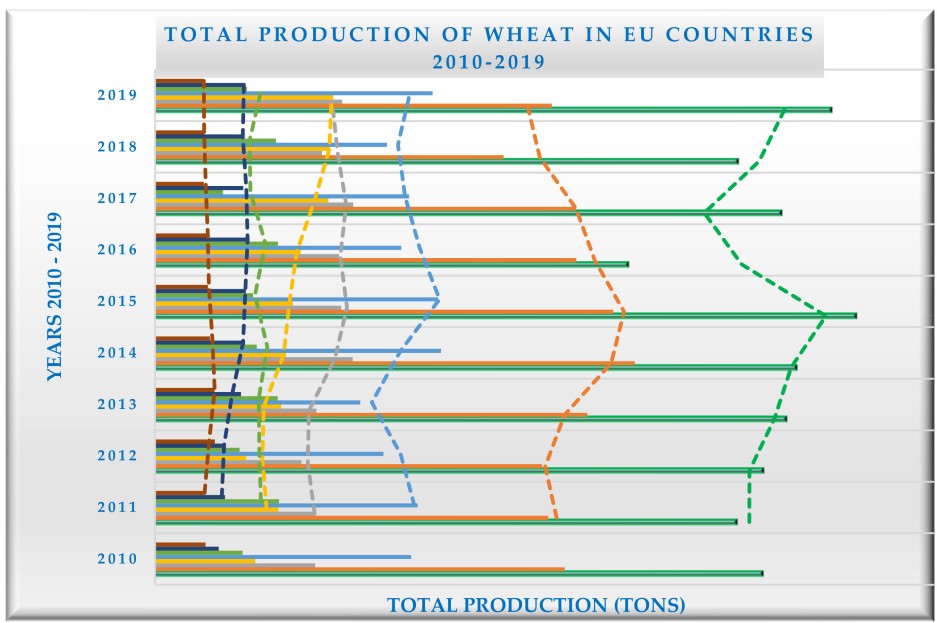

**Figure 3.** The total production from 2010 to 2019 in European countries (source: Eurostat, 8 November 2021 and own processing).

The highest total production of common wheat and spelt in France was 40,468.54 thousand tons (2015), and the lowest was 27,299.4 thousand tons (2016). The highest production in Germany was 27,711.20 thousand tons (2014), and the lowest was 20,125.40 thousand tons (2018). The highest production was 11,448.66 thousand tons (2017), and the lowest was 8457.47 thousand tons (2012). The highest production in Romania was 10,280.58 thousand tons (2019), and the lowest was 5275.63 thousand tons (2012). The largest production in the UK was 16,509.45 thousand tons (2014), and the lowest was 11,851.69 thousand tons (2013). The highest production in Spain was 7156.51 thousand tons (2011), and the lowest was 3922.01 thousand tons (2017). The highest production in Hungary was 5453.30 thousand tons (2016), and the lowest was 3679.51 thousand tons (2010). The highest production in Italy was 3453.55 thousand tons (2012), and the lowest was 2785.80 thousand tons (2019).

The multi-year average for average production/ha of common wheat and spelt during the analyzed period was 3.88 tons/ha for Romania which ranked 7th (3.88 tons/ha) after the UK (7.85 tons/ha), Germany (7.58 tons/ha), France (7.04 tons/ha), Italy (5.38 tons/ha), Hungary (4.91 tons/ha) and Poland (4.45 tons/ha), Spain (3.55 tons/ha). Romania held 48.23% of the multi-year average recorded by the UK (ranked first) and 55.11% of the multi-year average recorded by France, according to average production/ha for common wheat and spelt (Figure 4).

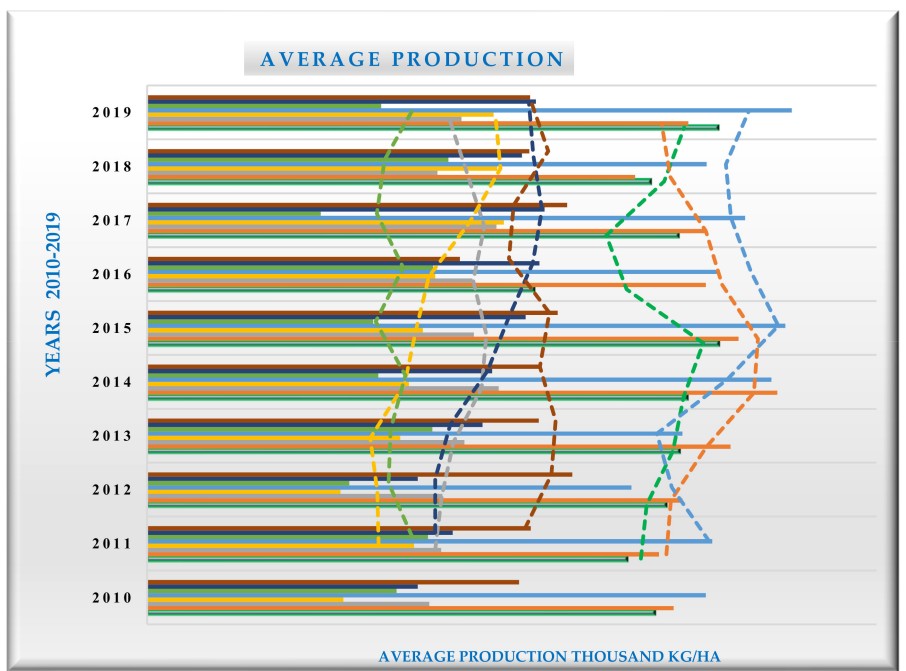

**Figure 4.** The multi-year average for average production/ha (source: Eurostat, 8 November 2021 and own processing).

The highest average yield/ha of common wheat and spelt in France was 7.83 thousand kg/ha (2019), and the lowest was 5.31 thousand kg/ha (2016). The highest average yield/ha in Germany was 8.11 thousand kg/ha (2015), and the lowest was 6.69 thousand kg/ha (2018). The highest average production/ha was 4.82 thousand kg/ha (2014), and the lowest was 3.87 thousand kg/ha (2010). The highest average production/ha in Romania was 4.89 thousand kg/ha (2017), and the lowest was 2.69 thousand kg/ha (2010). The highest average yield/ha in the UK was 8.84 thousand kg/ha (2019) and the lowest was 6.64 thousand kg/ha (2012). The highest average yield/ha in Spain was 4.13 thousand kg/ha (2018) and the lowest was 2.38 thousand kg/ha (2017). The highest average yield/ha in Hungary was 5.38 thousand kg/ha (2016) and the lowest was 3.71 thou-

sand kg/ha (2010 and 2012). The highest average yield/ha in Italy was 5.76 thousand kg/ha (2017) and the lowest was 4.29 thousand kg/ha (2016).

The following part of the study investigates the use of fertilizers and pesticides in agriculture in specified countries in order to determine their impact on wheat production.

Wheat culture, for optimal development, requires a specific consumption of chemical fertilizers, so as to reach the level of planned production, but the basic rule is that for one ton of wheat, 30–36 kg of fertilizers with N, 12 kg of fertilizers with $P_2O_5$ and 12 kg of fertilizers with $K_2O$. N is absorbed in nitric and ammoniacal forms, typically ending at flowering, but for larger harvests, it may continue until the grain-filling stage. The size of the harvest may depend on the continuation of N nutrition when $P_2O_5$ and $K_2O$ are present in sufficient quantities. Therefore, the determination and fractionation of N doses must consider the needs of the wheat plants during each growth phase, the amount of nitrogen in the soil that is accessible to the plants throughout the growing season, the mobility of N in the soil and the risk of its movement in depth with water from precipitation [30]. A well-developed root system adapted to the specificity of the soil positively influences the yield of the wheat crop [31].

The maximum consumption of nutrients in the wheat crop takes place in a short time, from the beginning of straw formation to ripening in the milk phase [30]. In wheat, embryonic roots grow and branch, providing the plant with water and nutrients. In addition, several adventitious or coronal roots emerge from the basal nodes of the stem, forming a fasciculate root system that is longer, vigorous, and heavily branched. Approximately two-thirds of the root mass reaches a depth of 25–30 cm, while only a small portion penetrates deeper into the soil [32]. The properties of the soil, such as texture, structure, capacity, moisture, and fertility, greatly influence cereal root development. To examine the relationship between the use of fertilizers and plant protection products and wheat production, this study determined the multi-year averages for the quantities of these substances used per hectare in the analyzed countries from 2010 to 2019. The study also highlighted the amount (in kg) of these substances used to produce one ton of common wheat and spelt.

Regarding N fertilizers, in the wheat crop, the optimal quantities ensure vigorous rooting and twining of the plants, good winter resistance, ears with many grains and rich in proteins. N deficiency slows down the growth of plants, causes their yellowing, poor resistance to adverse factors and decreases production. Excess N causes plants to grow excessively, to be prone to falling, to be prone to disease attacks and to be prone to the crushing of grains by delaying vegetation, thus reducing the effectiveness of applied fertilizers [33].

N fertilization is applied in variable doses, from 60 to 120 kg of active nutrients (a.n.)/ha. The N dose depends on the preceding crop, the application of manure, pedoclimatic conditions, etc. The N dose is corrected in the spring, depending on the condition of the crop and the mineral N reserve in the soil profile, and the dose will increase by 15–20 kg N/ha, when the crop has a weak twinning, or the dose is reduced by the same amount when it is well developed. These doses are also corrected according to the degree of the water supply of the soil, being necessary to reduce in dry springs and increase when the precipitation is more abundant (5 kg of N are taken into account for every 10 mm deviation from the average of the area). The amounts of N consumed in each country to produce one ton of common wheat and spelt, as well as the multi-year averages of these consumption rates, are shown below: in Romania, the consumption of N fertilizers, at 10.1 kg per ton of wheat, is the lowest among the analyzed countries and is 12.3 kg lower than the highest consumption rate in Poland, the highest N fertilizer consumer among the studied countries for wheat cultivation (Figure 5).

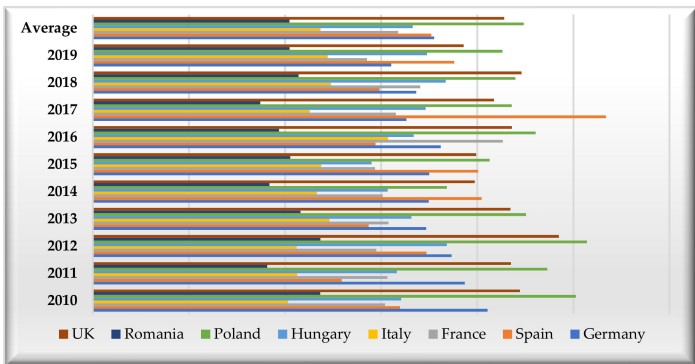

**Figure 5.** Annual consumption of N fertilizers (kg a.n.) to obtain one ton of wheat from 2010 to 2019, in the main European growing countries (Source: www.fao.org and own processing).

Poland is the largest consumer of N fertilizers in common wheat and spelt crops, with a multi-year average of 21.92 kg a.n. N consumed to obtain a ton of common wheat and spelt, and the deviation of the amounts of N used by Poland compared to those used in Romania (the lowest consumer of N fertilizers) is positive (+11.45 kg). The UK is the second largest consumer of N fertilizer, with a multi-year average of 19.70 kg N used and the deviation of the amounts of N used by the UK compared to the lowest consumer of N fertilizers (Romania) is positive (+7.23 kg). The multi-year average of 17.82 kg of N consumed placed Germany in third place among the analyzed countries, with a positive deviation of 7.23 kg compared to Romania, the smallest N fertilizer consumer. In Spain, the multi-year average of 16.82 kg of N consumed places it in fourth place, with a positive deviation of 6.35 kg compared to Romania. The multi-year average of 16.42 kg of N consumed places Italy in fifth place, with a positive deviation of 5.95 kg compared to Romania. The multi-year average in France is 15.88 kg of N consumed, placing it in sixth position, with a positive deviation of 5.41 kg compared to Romania. Italy is the seventh consuming country of N fertilizers and the multi-year average of N consumed in Italy is 11.67 kg, placing it in seventh position, with a positive deviation of 1.22 kg compared to Romania. Romania is the last country in the consumption of N fertilizers and the multi-year average of N consumed is 10.47 kg, placing it in the last position in the hierarchy, with a negative deviation of 11.45 kg compared to Poland (Figure 5).

The following section presents the main statistical indicators calculated for the amounts of N fertilizers (in kg) and their corresponding wheat productions (Figure 6).

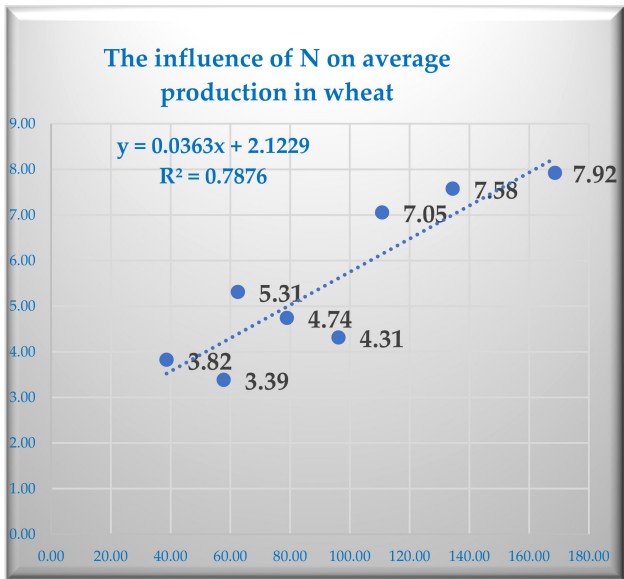

**Figure 6.** The influence of N on average wheat production. (Source: www.fao.org and own processing).

The main statistical indicators calculated for the amounts of N fertilizers (a.n. N) and related wheat productions will be presented next (Table 2).

**Table 2.** Standard deviation and coefficient of variation for average multi-year wheat production and the influence of fertilizers and pesticides (source: www.fao.org and own processing).

| Country | The Influence of Fertilizers and Pesticides on Average Wheat Production | | |
| :---: | :---: | :---: | :---: |
| | Multiannual Average (2010–2019) of Average Wheat Production —Tons/ha | Deviation Standard | The Coefficient of Variation |
| Germany | 7.58 | 0.57 | 7.51 |
| Spain | 3.39 | 0.57 | 16.71 |
| France | 7.05 | 0.73 | 10.31 |
| Italy | 5.31 | 0.43 | 8.14 |
| Hungary | 4.74 | 0.67 | 14.18 |
| Poland | 4.31 | 0.33 | 7.70 |
| Romania | 3.82 | 0.81 | 21.06 |
| UK | 7.92 | 0.68 | 8.59 |

We applied the Fisher test to the model for the relationship between average wheat production and the amount of N. F calc = 22.48 > F table = 5.32, with a confidence level of $\alpha = 0.05$ and df1 = 1, df2 = 8. Because F calc > F table, we accept hypothesis H1, which states that the model is plausible (Figure 6).

The average yields of the wheat crop were significantly influenced by the amounts of N used, namely 78.76% of the variation in production is due to the use of N in the countries studied (Table 3).

**Table 3.** The multiannual Average of Wheat Production, 2010–2019 (source: www.fao.org and own processing).

| Country | Multiannual Average (2010–2019) —kg a.n. N/Ton of Wheat | Multiannual Average (2010–2019) of Average Wheat Production —Ton/ha |
| :---: | :---: | :---: |
| Germany | 134.42 | 7.58 |
| Spain | 57.81 | 3.39 |
| France | 110.80 | 7.05 |
| Italy | 62.54 | 5.31 |
| Hungary | 78.93 | 4.74 |
| Poland | 96.24 | 4.31 |
| Romania | 38.65 | 3.82 |
| UK | 168.69 | 7.92 |

In common wheat and spelt crops, $P_2O_5$ fertilizers help plant rooting, increasing N efficiency, but depending on the soil, it is beneficial even if applied alone. If in the first phases of vegetation, $P_2O_5$ is quickly absorbed by the plants from the applied fertilizers, later they have the ability to use the $P_2O_5$ from the soil reserve. If the $P_2O_5$ doses are high, this aspect can cause the starch content to increase to the detriment of the grain protein content. Delayed growth of wheat plants with a weak root system as well as delayed maturity is caused by $P_2O_5$ deficiency. The quantity of easily soluble phosphates in the

soil is what determines $P_2O_5$ fertilization, and for an optimum, this content must not be less than 3.0–3.5 mg mobile $P_2O_5$/100 g of soil [13]. Agrochemical mapping of the soil is necessary for this purpose, but, as a general rule, the optimal economic dose of $P_2O_5$ (Table 4) is corrected according to the quantities of $P_2O_5$ fertilizers or manure applied in previous years.

**Table 4.** The specific consumption of $P_2O_5$ fertilizers in the wheat crop according to soil fertility.

| Phosphate Fertility of the Soil -Mg $P_2O_5$/100 g Soil- | Dose $P_2O_5$ -kg/ha- |
|---|---|
| Soils with high phosphate fertility (>5.5–7.5 mg $P_2O_5$/100 g soil) -fertilized soils in previous years with phosphate fertilizers and manure | 40–60 |
| Soils with medium phosphate fertility (3.5–5.5 mg $P_2O_5$/100 g soil) | 60–80 |
| Soils with low phosphate fertility (<3.5 mg $P_2O_5$/100 g soil) | 80–100 |

Source: [14].

It is necessary to specify that average wheat yields directly influence the profitability of farmers active in the agricultural sector in each country under study [14].

The amounts of $P_2O_5$ consumed in each country to produce one ton of common wheat and spelt, as well as the multi-year averages of these consumption rates, are shown below: Poland is the largest consumer of $P_2O_5$ fertilizers in common wheat and spelt cultivation, with a multi-year average consumption of 7.01 kg per ton of wheat, and a positive deviation of 4.17 kg compared to Germany. Spain is the second largest consumer of $P_2O_5$ fertilizers in common wheat and spelt cultivation, with a multi-year average consumption of 6.95 kg per ton of wheat, and a positive deviation of 4.11 kg compared to Germany. Romania ranks third in terms of $P_2O_5$ fertilizer consumption, with a multi-year average of 4.08 kg consumed per ton of wheat. This represents a negative deviation of 2.93 kg compared to Poland, the highest $P_2O_5$ fertilizer consumer, but a positive deviation of 1.24 kg compared to Germany, the lowest $P_2O_5$ fertilizer consumer (Figure 7).

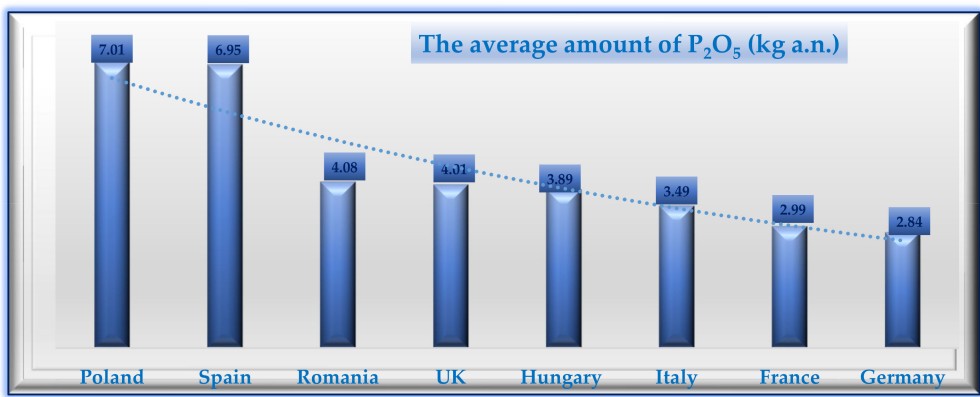

**Figure 7.** Common wheat and spelt crop—multiannual average amount of $P_2O_5$ (kg a.n.) (source: www.fao.org and own processing).

The UK is the fourth largest user of $P_2O_5$ fertilizers for common wheat and spelt cultivation, with a multi-year average consumption of 4.01 kg per ton of wheat, and a positive deviation of 1.18 kg compared to Germany. Hungary is the fifth largest consumer of $P_2O_5$ fertilizers for common wheat and spelt, with a multi-year average consumption of 3.89 kg per ton of wheat, and a positive deviation of 1.05 kg compared to Germany. Italy is the sixth largest consumer of $P_2O_5$ fertilizers for common wheat and spelt, with a multi-year average consumption of 3.49 kg per ton of wheat, and a positive deviation of

0.65 kg compared to Germany. France is the seventh largest consumer of $P_2O_5$ fertilizers for common wheat and spelt, with a multi-year average consumption of 2.99 kg/ton of wheat, and a positive deviation of 0.15 kg compared to Romania.

We find that the last position in terms of $P_2O_5$ fertilizer consumption for common wheat and spelt culture is occupied by Germany (multiannual average of 2.84 kg $P_2O_5$ consumed to obtain one ton of common wheat and spelt).

The main statistical indicators for the quantities of $P_2O_5$ fertilizers (a.n. $P_2O_5$) and the corresponding wheat productions are presented below (Table 5).

**Table 5.** Standard deviation and coefficient of variation for the multi-year average $P_2O_5$ quantity used to obtain one ton of wheat (source: www.fao.org and own processing).

| Specification | Phosphorous Fertilizers (a.n. $P_2O_5$) | | |
| --- | --- | --- | --- |
| | Multiannual Average —kg a.n. $P_2O_5$/Ton of Wheat | Deviation Standard | The Coefficient of Variation |
| Germany | 21.46 | 2.90 | 13.52 |
| Spain | 23.89 | 2.62 | 10.97 |
| France | 20.87 | 4.05 | 19.41 |
| Italy | 18.62 | 1.16 | 6.23 |
| Hungary | 18.46 | 6.27 | 33.94 |
| Poland | 30.92 | 2.56 | 8.27 |
| Romania | 15.04 | 3.38 | 22.46 |
| UK | 31.50 | 0.87 | 2.77 |

From the statistical analysis of the data, it is noted that Romania and Hungary have the highest values for the coefficients of variability V of the amounts of $P_2O_5$ in the 10 years.

The standard deviation and the coefficient of variation for the multiannual average wheat production under the influence of $P_2O_5$ are presented in Table 6.

**Table 6.** The multiannual Average of wheat production (source: www.fao.org and own processing).

| Country | Multiannual Average (2010–2019) —kg a.n. $P_2O_5$/Ton of Wheat | Multiannual Average (2010–2019) of Average Wheat Production —Ton/ha - |
| --- | --- | --- |
| Germany | 21.46 | 7.58 |
| Spain | 23.89 | 3.39 |
| France | 20.87 | 7.05 |
| Italy | 18.62 | 5.31 |
| Hungary | 18.46 | 4.74 |
| Poland | 30.92 | 4.31 |
| Romania | 15.04 | 3.82 |
| UK | 31.50 | 7.92 |

The calculated value of the Fisher test F = 0.51 is lower than the table value F table = 5.32, so we accept hypothesis H0, namely, that the model is not plausible.

The average yields of the wheat crop were very little influenced by the amounts of $P_2O_5$ used, namely 7.83% of the variation in production is due to the use of $P_2O_5$ in the countries included in the study (Figure 8).

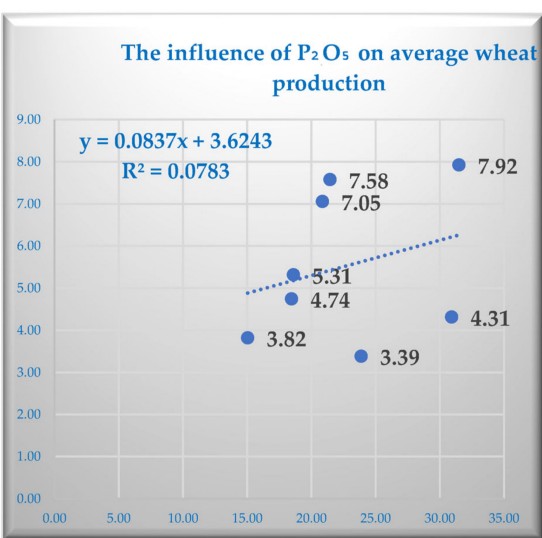

**Figure 8.** The Influence of $P_2O_5$ (source: www.fao.org and own processing).

Applied to favor the synthesis of carbohydrates and increased resistance to frost, wintering, and fall diseases, $K_2O$ fertilizers in insufficient quantities lead to the specific yellowing of the leaf blade in the upper part and on the edge of the leaf, and over time, the plant remains small and undeveloped, but the formation of numerous axillary branches at the base of the plant is also observed, and the plant eventually becomes a bush. S lightly eroded soils require $K_2O$ fertilization and that have a $K_2O$ content below 15 mg $K_2O$ accessible/100 g soil (the application involves doses of 50–60 kg/ha a.n).

Average yields for the wheat crop, in addition to being influenced by the applied fertilizers, are also influenced by crop rotation, according to specialists in the field [28].

In terms of $K_2O$ fertilizers consumption per ton of wheat, Romania ranks last, with a multi-year average consumption of 1.41 kg. This represents a negative deviation of 8.89 kg compared to Poland, the highest consumer among the studied countries. Poland is the largest consumer of $K_2O$ fertilizers in the cultivation of common wheat and spelt, with a multi-year average consumption of 10.08 kg, and a positive deviation of 8.62 kg compared to Romania (Figure 9).

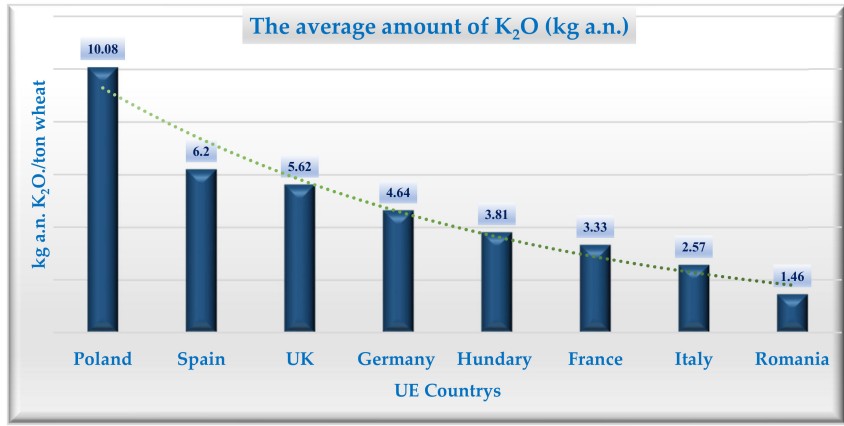

**Figure 9.** Average amount of $K_2O$ (kg a.n. $K_2O$) used to obtain a ton of common wheat and spelt (source: www.fao.org and own processing).

Spain is the second country consuming $K_2O$ fertilizers in common wheat and spelt crops (6.20 kg $K_2O$ multiannual average), and the deviation compared to Romania is positive (+4.74 kg a.n. $K_2O$). The UK is the third largest consumer of $K_2O$ fertilizers with a multi-year average of 5.62 kg a.n. $K_2O$ consumed to produce one ton of common wheat

and spelt. The deviation of the UK from Romania in terms of the consumption of $K_2O$ fertilizers is positive (+4.16 kg a.n.). Germany is the fourth consuming country of $K_2O$ fertilizers, with a multi-year average of 4.64 kg a.n. $K_2O$ consumed. Germany's deviation from Romania in terms of $K_2O$ fertilizer consumption is positive (+3.18 kg a.n.).

Hungary is the fifth country consuming $K_2O$ fertilizers with a multi-year average of 3.81 kg a.n. $K_2O$ consumed to produce one ton of common wheat and spelt. Hungary's deviation from Romania in terms of $K_2O$ fertilizer is positive (+0.11 kg a.n.). France is the sixth country consuming $K_2O$ fertilizers with a multi-year average of 3.33 kg a.n. $K_2O$ consumed. France's deviation from Romania in terms of $K_2O$ fertilizer is positive (+1.87 kg a.n.). Italy is the seventh country consuming $K_2O$ fertilizers with a multi-year average of 2.57 kg a.n. $K_2O$ consumed. Italy's deviation from Romania in terms of $K_2O$ fertilizer consumption is positive (+1.11 kg a.n.).

The main statistical indicators for the amounts of $K_2O$ fertilizers (a.n. $K_2O$) and the corresponding wheat productions from 2010 to 2019 related to the countries studied will be presented next (Table 7).

**Table 7.** The multiannual average $K_2O$ quantity used to obtain one ton of wheat. (Source: www.fao.org and own processing).

| Country | $K_2O$ Fertilizers (a.n. $K_2O$) | | |
| --- | --- | --- | --- |
| | Multiannual Average —kg a.n. $K_2O$/Ton of Wheat | Deviation Standard | The Coefficient of Variation |
| Germany | 35.02 | 2.06 | 5.89 |
| Spain | 21.33 | 1.95 | 9.14 |
| France | 23.26 | 1.68 | 7.21 |
| Italy | 13.72 | 3.19 | 23.24 |
| Hungary | 18.34 | 4.96 | 27.06 |
| Poland | 44.43 | 5.77 | 12.99 |
| Romania | 5.38 | 1.97 | 36.55 |
| UK | 44.13 | 1.67 | 3.78 |

In the case of $K_2O$, the calculated value of the Fisher test, F = 2.07 is lower than the table value F table = 5.32, so we accept the H0 hypothesis, namely that the model is not plausible (Figure 10).

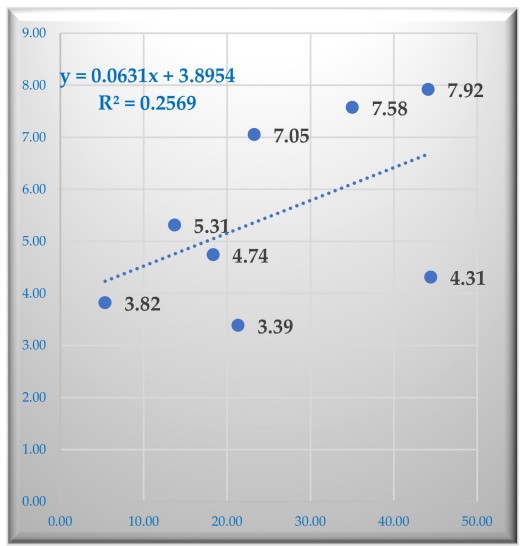

**Figure 10.** The influence of $K_2O$ on average wheat production (source: www.fao.org and own processing).

The standard deviation and the coefficient of variation for the multiannual average wheat production under the influence of fertilizers are presented in Table 8.

**Table 8.** The influence of $K_2O$ on average wheat production (source: www.fao.org and own processing).

| Country | Multiannual Average (2010–2019) —kg a.n. $K_2O$/Ton of Wheat | Multiannual Average (2010–2019) of Average Wheat Production —Ton/ha |
|---|---|---|
| Germany | 35.02 | 7.58 |
| Spain | 21.33 | 3.39 |
| France | 23.26 | 7.05 |
| Italy | 13.72 | 5.31 |
| Hungary | 18.34 | 4.74 |
| Poland | 44.43 | 4.31 |
| Romania | 5.38 | 3.82 |
| UK | 44.13 | 7.92 |

Average yields of the wheat crop were little influenced by the amounts of $K_2O$ fertilizer used, namely 25.69% of the yield variation is due to the use of $P_2O_5$ in the countries studied.

Integrated control is necessary to combat the attack of pathogens and pests on the wheat crop, avoiding the placement of wheat after grassy precursor plants, where a high degree of pathogens and pests (wireworm and scurvy beetle) which can be transmitted through seeds has been found (common downy mildew, downy mildew and fusarium wilt).

It is recommended to frame sowing in the optimal period, avoiding early sowing, balanced fertilization with NPK, the use of resistant varieties, compliance with the density of 450–550 germinable grains/$m^2$, and technological elements that contribute to reducing the attack of diseases and pests. The treatment of wheat seeds before sowing is carried out preventively to protect the plants from pathogens that are transmitted through the seed, with spores on the integument of the grain (common mildew, Fusarium wilt) or with spores inside the grain (flying smut). Some lands are more exposed to the attack of pests in autumn, in which case the use of insecticides is recommended. Especially in the case of monocultures, which favor the accumulation of large reserves of harmful organisms in the soil, the optimal treatment method includes the application of insect-fungicide products, which prevent both the appearance of diseases and the attack of pests.

Weed control is the main care work for the wheat crop, as crop losses caused by weed growth are between 10 and 70%. Dicotyledonous weeds are the ones that cause the greatest damage, and their control is mandatory. Monocotyledonous weeds are a problem for wheat crops in hilly, high-moisture areas.

Fighting wheat diseases is effectively achieved by combining preventive methods (cultivation of resistant varieties, compliance with rotation, ensuring normal density and balanced fertilization with NPK), with curative ones. The damage caused by various pathogens on the foliar apparatus of wheat reduces the assimilation capacity of the plants and causes production losses (about 10%) or a decrease in the quality of the obtained wheat [30].

Romania ranks last in the consumption of pesticides for wheat cultivation (0.19 kg a.n. pesticides used to obtain one ton of wheat), with a negative deviation (−1.03 kg a.n.) compared to Italy, which is the largest consumer of pesticides-total among the countries studied (Figure 11).

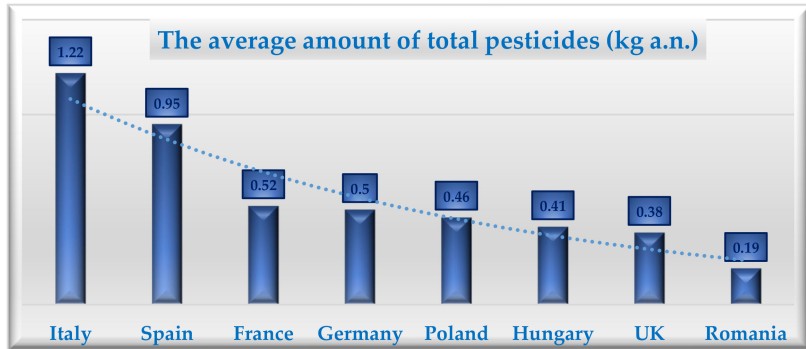

**Figure 11.** Wheat crop—the average amount of total pesticides (kg) used to produce one ton of common wheat and spelt (source: www.fao.org and own processing).

Italy is the first large consumer country of pesticides-total for common wheat and spelt crops (1.22 kg a.n. pesticides-total consumed to obtain one ton of common wheat and spelt). Italy's deviation from Romania in terms of total pesticide consumption is positive (+1.03 kg a.n.). Spain is the second country consuming pesticides-total in common wheat and spelt crops (0.95 kg a.n. pesticides-total consumed to obtain one ton of common wheat and spelt). Spain's deviation from Romania is positive (+0.76 kg a.n.) France is the third country consuming pesticides-total for common wheat and spelt crops (0.52 kg a.n. pesticides-total consumed to obtain one ton of common wheat and spelt). France's deviation from Romania is positive (+0.33 kg a.n.) Germany is the fourth country consuming pesticides-total in common wheat and spelt crops (0.50 kg a.n. pesticides-total consumed to obtain one ton of common wheat and spelt). Germany's deviation from Romania is positive (+0.31 kg a.n.) Poland is the fifth country consuming pesticides-total in the cultivation of common wheat and spelt (0.46 kg of pesticides-total consumed to obtain one ton of common wheat and spelt). Poland's deviation from Romania is positive (+0.23 kg a.n.) in terms of total pesticide consumption. Hungary is the sixth country consuming pesticides-total in common wheat and spelt crops (0.41 kg a.n. pesticides-total consumed to obtain one ton of common wheat and spelt). Hungary's deviation from Romania is positive (+0.22 kg a.n.). The UK is the seventh largest consumer of total pesticides in common wheat and spelt (0.38 kg of total pesticides consumed to produce one ton of common wheat and spelt). The UK's deviation from Romania is positive (+0.19 kg a. n.) (Figure 11).

Next, statistical indicators will be presented for the relationship between the number of pesticides used and wheat production (Table 9).

**Table 9.** Standard deviation and coefficient of variation for the multi-year average pesticide quantity used to obtain one ton of wheat (source: www.fao.org and own processing).

| Country | Pesticides-Total (a.n. Pesticides-Total) | | |
|---|---|---|---|
| | Multiannual Average —kg a.n. Pesticides-Total/Ton of Wheat | Deviation Standard | The Coefficient of Variation |
| Germania | 3.77 | 0.19 | 5.11 |
| Spain | 3.28 | 0.46 | 14.03 |
| France | 3.67 | 0.47 | 12.74 |
| Italy | 6.50 | 0.73 | 11.17 |
| Hungary | 1.96 | 0.20 | 10.01 |
| Poland | 2.03 | 0.15 | 7.35 |
| Romania | 0.69 | 0.07 | 10.61 |
| UK | 2.97 | 0.19 | 6.24 |

The standard deviation and the coefficient of variation for the multiannual average wheat production under the influence of total pesticides are presented in Table 10.

**Table 10.** The multiannual average of pesticides on average wheat production (source: www.fao.org and own processing).

| Country | Multiannual Average (2010–2019) —kg a.n. Pesticides-Total/Ton of Wheat | Multiannual Average (2010–2019) of Average Wheat Production —Ton/ha |
|---|---|---|
| Germany | 3.77 | 7.576 |
| Spain | 3.28 | 3.386 |
| France | 3.67 | 7.053 |
| Italy | 6.50 | 5.313 |
| Hungary | 1.96 | 4.743 |
| Poland | 2.03 | 4.314 |
| Romania | 0.69 | 3.823 |
| UK | 2.97 | 7.922 |

The average yields of the wheat crop were slightly influenced by the total amount of pesticides used, i.e., 11.17% of the variation in production is due to the use of total pesticides in the countries studied (Figure 12).

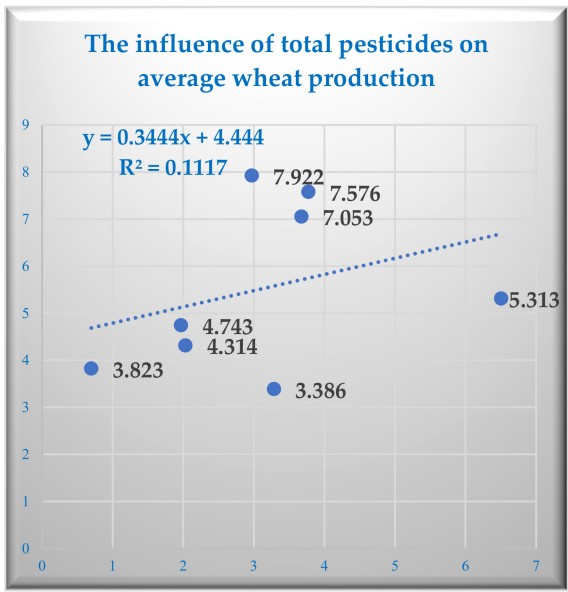

**Figure 12.** The influence of pesticides on average wheat production (source: www.fao.org and own processing).

The analysis of the correlation between the annual production of wheat in each country and the amount of N, $P_2O_5$, and $K_2O$, respectively, and pesticides allocated, from 2010 to 2019, by determining the Pearson correlation coefficient "r" in the countries studied, is presented in the following table (Table 11).

**Table 11.** Pearson correlation coefficient r (source: www.fao.org and own processing).

| Specification | The Pearson R Correlation Coefficient between the Annual Production of Wheat and the Amount of N, $P_2O_5$, $K_2O$, Respectively, and Pesticides Allocated from 2010 to 2019 | | | | | | | |
|---|---|---|---|---|---|---|---|---|
| | Germany | Spain | France | Italy | Hungary | Poland | Romania | UK |
| N | 0.59 | −0.17 | 0.15 | 0.07 | 0.88 | 0.06 | 0.88 | 0.77 |
| $P_2O_5$ | 0.59 | −0.02 | 0.31 | 0.07 | 0.89 | −0.79 | 0.82 | 0.51 |
| $K_2O$ | 0.49 | 0.14 | 0.82 | −0.20 | 0.85 | 0.44 | 0.69 | 0.60 |
| Pesticides | 0.34 | 0.06 | 0.16 | −0.09 | 0.15 | 0.77 | −0.47 | 0.49 |

We consider that they are significant, following the table with significance thresholds for the "r" coefficient for df = 8 degrees of freedom, values greater than 0.632.

The analysis of the values obtained for the Pearson correlation coefficient "r" in the countries studied highlighted the following:

- In Spain and Italy, the correlation between the amounts of N, $P_2O_5$, and $K_2O$ and the average wheat production is insignificant, with the "r" coefficient having values close to 0.
- Romania and Hungary show a very high direct correlation between the number of fertilizers used and the average wheat production (r > 0.632). On the other hand, the link between average production and pesticides is negative and not very significantly high in Romania (r = −0.47), which is considered an almost non-existent link in Hungary (r = 0.15 < 0.632).
- In Germany, the correlation is direct and not intense for all types of fertilizers, and pesticides have a lower influence on average wheat production (r < 0.632).
- Taking into account the synergy of the action of fertilizers and plant protection products, son the average wheat production, it is found that it is strongly correlated with the amounts of N used in countries such as Romania, Hungary and the UK, a conclusion highlighted by the value close to one for r.

Multiple linear regression analysis was also conducted to examine the relationship between wheat production and the use of fertilizers. The correlogram indicates that there is a strong direct correlation between wheat production and the amount of N applied.

We note the average wheat production, which is the explained variable with Y. The explanatory variables, which are the amounts of $K_2O$, N and $P_2O_5$, are noted with X1, X2, and X3, respectively (Figure 13). The multiple linear regression model obtained is the following:

$$Y = 3.549 - 0.036 \times X1 + 0.056 \times X2 - 0.098 \times X3 \tag{1}$$

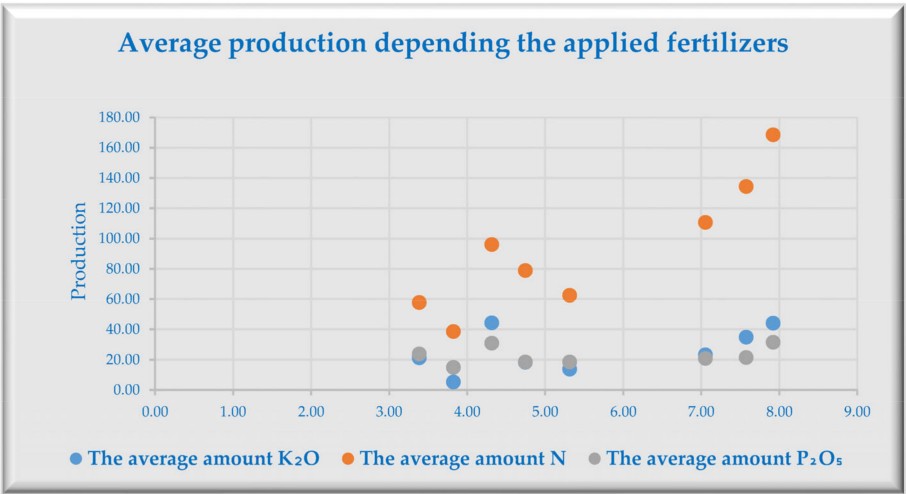

**Figure 13.** Average wheat production depends on the applied fertilizers (source: own processing).

The model's validity is supported by the values of the coefficient of determination $R^2 = 0.95$ and the Adjusted R Square = 0.91, both of which are close to 1. This indicates that the model can be effectively used in economic analyses.

$$y = 3.169 + 0.177 \times x1 - 0.025 \times x2 - 0.106 \times x3 + 0.052 \times x4 \tag{2}$$

where y is the dependent variable (average wheat production) and x1 and x4 are the independent variables. Notably, x1—pesticides used avrg kg/ha, x2—$K_2O$ avrg kg/to, x3—$P_2O_5$ avrg kg/to, x4—N avrg kg/to.

We can use the R-Squared value to understand the relationship between the independent variables. A high R-Squared value (close to 1) indicates that a large percentage of the variance in the dependent variable is explained by the independent variables. In our multiple regression, R Square is equal to 0.978, which means that a large percentage of the variance in the dependent variable is explained by the independent variables.

Only the *p*-value for x4 (N) is less than 0.05 which is used to test the hypothesis that the coefficients in the regression equation are equal to zero. In our case, only the *p*-value for x4 is less than the significance level, so only that coefficient is significantly different from zero and has an effect on the dependent variable (wheat production average). The t stat is used to test if the coefficient is equal to zero. A high t-statistic which resulted only for the coefficient of x4 (N) indicates that the coefficient is significantly different from zero.

This means that for every 1 unit change in x4, the dependent variable will be expected to change on average with 0.052 units, provided that the other variables remain constant.

For the rest of the independent variables (x1, x2, x3) the *p*-value is not less than 0.05, so the coefficients are significantly different from zero.

Currently, according to studies in the field, there are a number of solutions that contribute to reducing the amounts of fertilizers and pesticides used in wheat cultivation, and not only, so as to reduce, on the one hand, the negative impact on people's health, and on the other hand, to reduce the degree of pollution. One such solution is single fertilization with a controlled-release fertilizer in the winter wheat crop. It should also be remembered that this solution contributes to reducing labor costs, but also to increasing N use efficiency [34]. Another practice proposed by specialists that must be taken into account is the application to the wheat crop of differentiated doses of N fertilizer with different application times. This fact would lead both to an increase in production and an optimization of resources for a certain agroecological area [35].

In the world agricultural economy, wheat production ranks third, after corn and rice. Since, at the present time, a significant part of wheat production has multiple uses (food for the population, animal feed, industry), even for obtaining biofuels that are important for the contemporary economy, greater attention is required regarding wheat cultivation, simultaneously with the reduction of pesticides and fertilizers [36]. Otherwise, a sustainable policy for the use of pesticides and fertilizers is needed in the medium and long term.

### 4. Conclusions

Based on the studies carried out and the bibliography analyzed, it is found that N fertilizers decisively influence the production of wheat, those with $P_2O_5$ and $K_2O$ bringing a significant contribution to the production when they are applied within the N fertilizer technology. The use of pesticides is important for increasing the quantity and quality of wheat production, but it is essential to use them in a way that does not negatively impact soil fertility and the environment.

The research carried out on the wheat crop regarding the influence that fertilizers and pesticides have on production highlighted the following:

1. The multiannual average area cultivated with common wheat and spelt in the countries studied varied between 547.58 thousand ha and 4924 thousand ha.
2. The total multi-year wheat crop production in the countries under study varied between 2994.3 thousand tons and 34,790.95 thousand tons.

3. Average wheat production/ha ranges from 3.55 tons/ha to 7.85 tons/ha, with the UK having the highest average production/ha.
4. Regarding the consumption of fertilizers and pesticides, the following was found:
    ○ The average yields of the wheat crop were significantly influenced by the amounts of N used, i.e., 78.76% of the variation in production is due to the use of N in the countries included in the study. Poland has the highest consumption of N fertilizers for wheat production at an average of 21.92 kg per ton, while Romania has the lowest consumption at 10.1 kg per ton.
    ○ The average productions of the wheat crop were very little influenced by the amounts of $P_2O_5$ used, namely 7.83% of the production variation is due to the use of $P_2O_5$ in the countries analyzed in this study. From the statistical analysis of the data, Romania and Hungary stand out as the countries that have the highest values for the coefficients of variability V of $P_2O_5$ amounts in the 10 years analyzed. Poland is the largest consumer of $P_2O_5$ fertilizers for common wheat and spelt (7.01 kg $P_2O_5$ consumed to obtain one ton of common wheat and spelt), and Germany is in the last position in terms of fertilizer consumption with $P_2O_5$ in the common wheat and spelt crop (2.84 kg $P_2O_5$ consumed to obtain one ton of common wheat and spelt).
    ○ Average yields of the wheat crop were little influenced by the amounts of $K_2O$ used, namely 25.69% of the variation in production is due to the use of $P_2O_5$ in the countries included in the study. Poland is the largest consumer of $K_2O$ fertilizers in the common wheat and spelt crop (10.08 kg $K_2O$ consumed to obtain one ton of common wheat and spelt). Romania is positioned last in terms of $K_2O$ consumption, with a quantity of 1.41 kg a.n. of $K_2O$ used to obtain a ton of wheat.
    ○ The average yields of the wheat crop were weakly influenced by the amounts of pesticides-total used, i.e., 11.17% of the variation in production is due to the use of pesticides-total in the countries included in the study. Italy is the first large consumer country of pesticides-total for common wheat and spelt crops (1.22 kg a.n. pesticides-total consumed to obtain a ton of common wheat and spelt) and Romania is positioned at the last place (0.19 kg a.n. pesticides-total used to obtain one ton of wheat).
    ○ The analysis of the values obtained for the Pearson correlation coefficient "r" in the countries included in the study, highlighted the following: in Spain and Italy the correlation between the amounts of N, $P_2O_5$, $K_2O$ and the average wheat production is insignificant, Romania and Hungary show a very direct correlation high between the number of fertilizers used and average yield in wheat, and the relationship between average yield and pesticides is negative and significantly high. In Germany, the correlation is direct and quite intense for all types of fertilizers, and pesticides have a lower influence on average wheat production.

Taking into account the synergy of the action of chemical fertilizers and pesticides-total on average production in wheat, it is found that it is strongly correlated with the amounts of N used in countries such as Romania, Hungary and the UK, a conclusion highlighted by the value close to one for "r".

By analyzing using a multiple regression model the multi-yearly average production and pesticides and fertilizers used in each country from the current study, it can be concluded that for every 1 unit change in the values of kg of N used per ha, the average wheat production per ha will be expected to change on average with 0.052 units, provided that the other variables remain constant (the rest of the fertilizers and the pesticides).

**Author Contributions:** Conceptualization, V.C.T., P.S. and T.A.D.; Methodology, V.C.T., P.S., T.A.D. and L.D.; Software, A.M.I., L.D. and E.A.D.; Validation, V.C.T. and A.M.I.; Formal analysis, I.-A.C., E.S., L.D. and D.I.S.; Investigation, V.C.T., P.S., M.M.M., D.I.S. and E.A.D.; Resources, V.C.T.,

A.M.I., M.M.M. and D.I.S.; Data curation, I.-A.C., M.M.M. and E.A.D.; Writing–original draft, P.S.; Visualization, E.S.; Supervision, T.A.D.; Project administration, P.S. All authors have read and agreed to the published version of the manuscript.

**Funding:** This research received no external funding.

**Institutional Review Board Statement:** Not applicable.

**Informed Consent Statement:** Not applicable.

**Data Availability Statement:** Not applicable.

**Acknowledgments:** We would like to thank the referees for all the data provided for this paper. The publication of this article was possible due to project no. 182/23.11.2021. The socio-economic impact of the application of the FARM TO FORK strategy in agriculture and its transposition in Romania contracted with Association of Maize Producers from Romania.

**Conflicts of Interest:** The authors declare no conflict of interest.

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
