# Peer review of "The Use of Fertilizers and Pesticides in Wheat Production in the Main European Countries"

_sustainability, doi:10.3390/su15043038_

Round 1

Reviewer 1 Report

Hello authors, an interesting documentation about impacts of use of fertilizers and pesticides on wheat production in the main EU growing countries. An issue that is simple in practice but should be examined routinely at certain periods. Research appears to have been carried out well and according to accepted protocols. However, it has some questions and corrections.

Title: Please correct it as “The impact of fertilizers and pesticides used on wheat production in the main EU growing countries”

Abstract: No corrections are required

Introduction: Line 72-79, this information must be cited and discussed from country and country. Additionally, medium or high yield should be defined. Other suggestions are in attached file.

Materials and methods: Some corrections are in attached file. Except for these, no corrections are required.

Results and discussions:

-          Spelled wheat has to be definitely defined. Detailed suggestion is in attached file.

-          Surface area is not necessary, only “area” is enough

-          Line 155, please move this sentence (in parentheses !!!) to end of the legend of Figure 2. Please apply this for all figures and tables.

-          Please use the standard font and size (indicating in writing rules in website of MDPI) in all figures and tables.

-          Please not use "bold" for country's name

-          Legends are not necessary in figure. Please delete them from all figures.

-          Line 272-275, N level (30-36 kg) is very high and P is not enough to reach one ton. Please discuss it in detail at the end of this part. This information should be moved after end of the regarding data and discussed. In addition, it must be cited.

-          Line 276-279, this information must be cited.

-          Please correct Fig. 5

-          Please use the standard font and size for all figures and tables.

-          Other corrections and suggestions are in the attached file.

-          The findings obtained from this study should be discussed with similar studies.

Reviewer 2 Report

This manuscript presents a Meta-analysis of the influence of fertilizer and pesticide usage on wheat production in major wheat-growing countries in the European Union. The results of the study indicate a fluctuation in wheat-cultivated areas and the average wheat production is affected by fertilizer and pesticide usage. A relationship between fertilizer usage and wheat production was evident for some countries, and not for others. Between the first two wheat-producing countries in the European Union France and Germany, wheat production was related to phosphatic fertilization (in terms of statistical significance) in France. However, for Germany all the correlations between wheat yield and types of fertilizers used were reported to be direct and quite intense it appears to be not so if the number of entries (n) is taken into consideration. In addition, there are certain shortfalls as mentioned below and on the annotated manuscript, which need attention.

1.      One of the major problems with studies like these is the assumption that the crop yield is related to the individual nutrient input. Contrarily, there is always an interactive effect of different nutrients on plant growth and yield (e.g. https://doi.org/10.1016/B978-0-12-811308-0.00004-1; https://doi.org/10.3389/fpls.2021.665583). The yield of a crop is therefore dependent on the balanced supply of different nutrients rather than an individual type of nutrient supplementation. The study should also focus on this line. I would suggest the authors perform a multiple regression analysis or a PCA involving all the fertilizer types and see how total fertilization affects wheat yield in different countries.

2.      Organic agriculture is gaining popularity in recent years and the area under organic agriculture for different crops including wheat is increasing globally. Provide information on the wheat area under organic cultivation in the studied countries. Did the data used in the study exclude wheat yield under organic cultivation or was it inclusive? If the yield data is inclusive of organic cultivation then the outcome could be misleading.

3.      Water and fertilizer demand for wheat can vary with the cultivated varieties/cultivars. Therefore, baseline information on the different wheat cultivars grown in the studied countries and their nutritional demand would reveal the extent of fertilizer use in these different countries. This is also true with the susceptibility of cultivated wheat cultivars to pests and pathogens.

4.      The abstract should be self-contained. Please check and revise the abstract in line with the revised statistical analysis. The conclusion(s) drawn from the results of the present study is missing.

5.      The objectives of the study are not clear. Specify what the results of the study on the use of synthetic fertilizers and pesticides on wheat productivity in different countries in the European Union would mean in the future. What about the proportion of fertilizer application to wheat yield in different countries? Can the use of synthetic chemicals be reduced if their use/utilization efficiency by the crop is increased? Indicate how the outcome of this study would contribute to the doubling of agricultural production in 2050 as mentioned in Line 111.

6.      The parametric Pearson’s correlation requires normal distribution of data. It is not clear if the data used in the analysis were checked for normality before use.

7.      The results observed are not adequately discussed. A wide fluctuation in the cultivable area and wheat yield was observed for different countries between 2010 and 2019. Nevertheless, the reasons for the fluctuations in these variables are obscure. Similarly, there is also some negative correlation between the use of certain fertilizers and wheat yield. For example, the wheat yield was inversely related to the use of phosphatic fertilizer use in Poland (Table 12). The possible reasons for a decline in wheat yield with increasing phosphatic fertilization need a critical argument.

Other comments:

8.      Line 44: Mention some of the characters that give cereals special value.

9.      Lines 69–71: But the negative influence of the sustained use of synthetic fertilizers on soil fertility should also be taken into consideration.

10.  Lines 72–74: Are the wheat-growing seasons uniform across all the wheat-growing European countries?

11.  Lines 118–122: Inferring the significance of the correlation based on the closeness of the correlation coefficient to a positive one or a negative one may be misleading. The strength of the correlation coefficient (significance) should be based both on the value of the correlation coefficient r and the sample size (n), together.

12.  Line 132: Is there any specific reason for not extending the study beyond 2019?

13.  Figure 3: Present the decimals uniformly across total production. The highest and lowest production country-wise could be highlighted in the data presented in the figure (possibly by different colours) so the portion Lines 222 to 237 can be removed.

14.  Figure 4: See comments for figure 3. 

15.  Lines 276–278: The statement is not correct. Wheat has a fibrous root system with long root hairs, which makes it more efficient in nutrient and water uptake. Moreover, the root system of wheat is produced in the soil profile where nutrient cycling is at the maximum (topsoil). For more details, consult https://doi.org/10.1007/s00122-021-03819-w.

16.  Line 447: Remove the space in the word slightly.

17.  Figure 6: Indicate the level of significance for the coefficient of determination.

18.  Figure 8: Delete the figure and indicate the results in the text. The coefficient of variation values appears to be very low and not significant.

19.  Figure 10: Indicate the level of significance for the coefficient of determination.

20.  Lines 525-535: Normalize. Is there any specific reason for presenting this portion in bold?

21.  Table 12: Indicate the variables involved in the correlation. Is it the relation between fertilizer use and wheat production? Clarify. Remove the column on the serial number. Indicate which of the correlation coefficients are significant values (I think it should be those values larger than 0.632 at P<0.05; when n-2 =8).

22.  Lines 594–608: The significance of correlation should be based both on the value of the correlation coefficient r and the sample size (n), together as mentioned earlier.

Round 2

Reviewer 1 Report

The authors have made corrections to a large extent. The abstract should be no more than 200 words in total, but the abstract is too long after corrections / additions. Therefore, it is necessary to shorten it.

Author Response

I reduced the summary according to your suggestions.
Thank you for your support!

Authors

Reviewer 2 Report

In the revised version the authors have taken into consideration all the suggestions raised in my previous review and revised the manuscript accordingly.

Author Response

Thank you for support!

Authors